# Species distribution of *Cannabis sativa*: Past, present and future

**Anna Halpin-McCormick**[1] **Tai McClellan Maaz**[1] **Michael B. Kantar**[1]* **Kasey E. Barton**[2] **Rishi R. Masalia**[3] **Nick Batora**[3] **Kerin Law**[3] **Eleanor J. Kuntz**[3]

**1** Department of Tropical Plant and Soil Sciences, University of Hawaii at Manoa, Honolulu, Hawaii, United States of America, **2** LeafWorks Inc, Sebastopol, California, United States of America, **3** Department of Life Sciences, University of Hawaii at Manoa, Honolulu, Hawaii, United States of America

* kant0063@umn.edu, mbkantar@hawaii.edu

## Abstract

*Cannabis sativa* L. is an annual flowering herb of Eurasian origin that has long been associated with humans. Domesticated independently at multiple locations at different times for different purposes (food, fiber, and medicine), these long-standing human associations have influenced its distribution. However, changing environmental conditions and climatic fluctuations have also contributed to the distribution of the species and define where it is optimally cultivated. Here we explore the shifts in distribution that *C. sativa* may have experienced in the past and explore the likely shifts in the future. Modeling under paleoclimatic scenarios shows niche expansion and contraction in Eurasia through the timepoints examined. Temperature and precipitation variables and soil variable data were combined for species distribution modeling in the present day and showed high and improved predictive ability together as opposed to when examined in isolation. The five most important variables explaining ~65% of the total variation were soil organic carbon content (ORCDRC), pH index measured in water solution (PHIHOX), annual mean temperature (BIO-1), mean temperature of the coldest quarter (BIO-11) and soil organic carbon density (OCDENS) (AUC = 0.934). Climate model projections where efforts are made to curb emissions (RCP45/SSP245) and the business as usual (RCP85/SSP585) models were evaluated. Under projected future climate scenarios, shifts worldwide are predicted with a loss of ~43% in suitability areas with scores above 0.4 observed by 2050 and continued but reduced rates of loss by 2070. Changes in habitat range have large implications for the conservation of wild relatives as well as for the cultivation of *Cannabis* as the industry moves toward outdoor cultivation practices.

## Introduction

*Cannabis sativa* is an annual diecious herb of Eurasian origin and inhabits a range of distinct geographies and climates [1–3]. These environments are characterized by having a good water supply and a range of soils that tend to be well-drained, nitrogen-rich, loamy and alluvial [4,5]. Humans have long had a relationship with *Cannabis* with the first observations of seeds associated with pottery fragments dated to ~10,000 years ago [4,6,7]. This long human use has made the taxonomy of the *Cannabis* genus a major question. Historically, it has been broadly

**Data availability statement:** All code and data is available at https://github.com/ahmccormick and high resolution figures are available at https://figshare.com/authors/Anna_H_McCormick/17741367 .

**Funding:** The author(s) received no specific funding for this work.

**Competing interests:** "LeafWorks is a for profit company that provided computing to help with this manuscript. This does not alter our adherence to PLOS ONE policies on sharing data and materials."

divided into two types, hemp- or drug-type, with early descriptions dating back to Linnaeus (1753) and Lamarck [4,8,9]. Linnaeus described the plants from Northern Europe as *Cannabis sativa* and Lamarck described plants from India as *Cannabis indica.* In 1924 an additional purported wild species growing in central Russia was described by Janischevsky [10] and termed *Cannabis ruderalis.* At the time, morphological differences between these three taxa led to the proposition of multiple species (sativa, indica and ruderalis) however, more recent work supports the rank of subspecies [11] with genetic diversity occurring across a latitudinal gradient along which classic differentiating phenotypes occur [12]. Due to the cross fertility of the proposed multispecies, *Cannabis* is now more commonly considered a monotypic genus [8,13,14]. The domestication of *Cannabis* occurred for fiber, seed and cannabinoid content [4,15,16]. The divergence of hemp and drug-type *Cannabis* ancestors from wild populations occurred ~ 12,000 years ago, followed by a separation of hemp and drug-type gene pools occurring ~ 4,000 years ago [16]. Hemp-type *Cannabis* can be further sub-classified into Narrow Leaf Hemp (NLH) or Broad Leaf Hemp (BLH) with all hemp types classified as *C. sativa* ssp *sativa.* Pollen grains in the archeological record has revealed that NLH spread out of the putative ancestral zone in Central to Northern Eurasia moving westward across into mainland Europe around 6 million years ago (MYA) whereas in contrast BLH spread into Southern China and Southeast Asia about 1.2 MYA [3,17]. Similarly, drug-type *Cannabis* can be divided into Narrow Leaf Drug (NLD) originating from South Asia (named "*sativa*" in the recreational market) and Broad Leaf Drug (BLD) originating from Central Asia (referred to as "*indica*" in the recreational market) [18]. The NLD varieties are typically found along the Himachal Pradesh with a range that expands into the Montagne regions of Northern India, Uttarakhand, Nepal, Sikkim, Bhutan and into the Arunachal Pradesh [3]. On the other hand, BLD varieties are reported to have spread from Jammu and Kashmir into Pakistan and Afghanistan [3]. *Cannabis* occurs in a range of diverse habitats, from cold and dry climates with short growing seasons in temperate regions to warm and wet climates with longer growing seasons in the tropics [4]. Since its divergence from *Humulus Lupulus* between 18.23 – 25.4 MYA [12,19], *Cannabis* has been exposed to many different environmental stressors. It has been hypothesized that following the last glacial maximum (LGM), *Cannabis* plants migrated into and persisted in refugia sites (e.g., Hengduan Mountains, Yungui Plateau, Caucasus Mountains, northern Mediterranean peninsula) until conditions were favorable once again for range expansion [4]. Habitat fragmentation and isolation driven by changing climate may have aided population separation and driven adaptive divergence among sub-species.

Currently, *Cannabis* cultivation occurs in both outdoor and indoor settings with hemp plants producing fibers and seed oils usually cultivated outdoors and high value medicinal and recreational drug-type plants often cultivated indoors or in more controlled farming systems [20,21]. Despite this widespread cultivation, no prior species distribution modeling for *Cannabis* has been published for present day or future scenarios. As outdoor cultivation becomes the standard practice, species distribution maps can help define and rank the importance of critical environmental properties and identify where these suitable environments may exist around the world currently and how they are likely to be affected by changing climate in the future. Further, there has been increased scientific interest in the last decade around understanding the soil determinants important for *Cannabis* cultivation [22].

Under climate change it is very possible that in the future, different regions may be more appropriate for different crops [23]. Species distribution models (SDMs) provides a method for informed land selection by identifying regions with favorable climatic and soil conditions and providing a suitability score for identifying these regions. SDMs can also support conservation decision making by identifying where other suitable habitats may be for endangered species and identifying the rank in important environmental factors involved in their

distribution. Additionally, shifting climate also requires long term planning to prepare for future changes in agricultural requirements for outdoor production. Currently, one major region for *Cannabis* cultivation is in California, in particular the Emerald Triangle. This state has become a major cultivation center in recent history and continues to be one of the largest *Cannabis* markets in the world with $5.3 billion in sales in 2022 [24]. However, it is currently unknown how future climate changes may impact *Cannabis* cultivation and conservation globally and what are the most important environmental variables that explain *Cannabis* species distribution. Therefore, the objectives of this study were twofold 1) integrate publicly available soil data and climate data, past, present, and future with wild *Cannabis* occurrence points to understand how regions of the world change in suitability for *Cannabis* through time and 2) specifically explore present-day and future suitability for *Cannabis* cultivation globally and in the state of California, as this state is known worldwide for its outdoor cultivation.

## Materials and methods

### Occurrence points

Occurrence data were obtained from iNaturalist (GBIF.org - https://doi.org/10.15468/dl.d8n6hx). This dataset contained 416 occurrence points which had paired images for each occurrence point (Table S1). Of these, 302 were deemed as wild or escapees and growing without human intervention (S2 Table). Plants were classified as wild if image inspection revealed no visible man-made objects of any kind, the plant was growing amongst other plants and the landscape appeared unmanaged. After removing duplicates there were 234 observations. After filtering for a longitude greater than zero, 137 observations remained and were used for SDM construction (S3 Table; S1 Fig). Eurasia is considered the center of origin of *Cannabis,* which provided rationale for including only data points with a longitude greater than zero in this study.

### Environmental variables

Occurrence points were used to query the datasets examined in this study which included the WorldClim 2.1 climate data (all 19 bioclim variables for temperature and precipitation, S4 Table) as well as monthly climate data for solar radiation, wind speed, water vapor pressure and elevation. Data was downloaded at the highest available spatial resolution of 30 seconds (~1 km$^2$) [25]. Soil properties were downloaded from the global soil database ISRIC World Soil [26] (S5 Table). Paleoclimate data were sourced from paleoclim.org with the highest spatial resolution of 2.5 arc-minutes (~5km) downloaded [27] (S6 Table). Future climate data were sourced from the WorldClim repository for SSP245 (mitigation) and SSP585 (business as usual) for 2050 (averages for 2041-2060) and 2070 (averages for 2061-2080) at a spatial resolution of 30-arc seconds. Future climate projections were based on the Intergovernmental Panel on Climate Change (IPCC) sixth assessment report (AR6) and used shared socioeconomic pathways (SSPs) for different global climate models (CMIP6). California soil orders were downloaded from https://databasin.org. Map overlays were created using the 'raster', 'rworldmap', 'ggplot2', 'sf' and 'mapdata' packages in RStudio [28].

### Model building

These bioclimatic and soil data were used to create species distribution models (SDM) using the software Maxent (Version 3.4.4 [29] in RStudio (Version 2022.2.0.443 [28]). Suitability maps were created using the Maxent software (Version 3.3.4). For historic distribution models, present day latitude and longitude data points (n = 137) were used under the assumption

that where *Cannabis* grows wild today may reflect where it could have grown in the past. Suitability maps were overlaid for the present day (1970-2000), 2050 and 2070, with a suitability cutoff score of 0.2. Acceptable suitability is defined as 0.2 for cultivated regions [30] and 0.4 for natural areas [31].

## Model evaluation

Model quality was explored using area under the curve (AUC) and the standard deviation of the AUC across replicates (SDAUC). A good model required an AUC ≥ 0.7 and SDAUC < 0.15. Shape files used for cropping raster extents included the World Administrative Boundaries, Countries and Territories shape file and the United States shapefiles (states_21basic). Asia and Russia and Europe and Southwest Asia shape file were retrieved from Stanford University EarthWorks (2008). Detailed World Polygons (LSIB) Europe and Southwest Asia, 2013. [Shapefile]. United States. Department of State. Office of the Geographer. Humanitarian Information Unit). For assessing habitat reduction, the 'rasterio; and 'numpy' packages were used in PyCharm (version 2023.3.3) to identify pixels above the threshold examined (above 0.4 suitability). These files were then imported into RStudio where pixel counts were converted to km² accounting for the curvature of the earth over the range of latitudes in each file. All code is available at https://github.com/ahmccormick and high resolution figures are available at https://figshare.com/authors/Anna_H_McCormick/17741367.

## Results and discussion

### Historic changes in the distribution of *Cannabis*

Eurasia is the putative ancestral zone for *Cannabis* and thus was the focus for the historical timepoints (S6 Table, S11 Fig). Across this geography there were many changes in the extent of suitable native habitat over geologic time. At 3.3 million years ago (mya) high potential suitability was observed in west and central Eurasia, as well as temperate South and East Asia (S11A Fig). At 3.2 million years ago during the Mid-Pliocene Warm Period (mPWP), high potential suitability remained in West-central Eurasia (S11B Fig), as well as temperate South and East Asia. There is also an increase in potential suitability in the Tibetan Plateau region (S11B Fig) as well as in Bangladesh and Northern Myanmar. The mPWP offers an opportunity to examine how a warmer than present world may have affected species distribution, as climate models estimates for this time show that the mean surface temperature globally was between 2.7- 4.0 °C higher than present along with high atmospheric $CO_2$ concentrations (350-450 ppm compared to 280 ppm pre-industrial revolution)[32]. Highlighting the significance of the increase in global surface temperature during the mPWP, annual temperature explained nearly 60% of the variation during the mPWP (S12B Fig). Between 3.2 million and 787,000-years ago there was a loss of lower suitability ranges (0.2 - 0.4) along the Tibetan Plateau. Habitat changes likely facilitated separation of Eurasian Steppe populations from those in China and the Himalayan Mountains (S11B and S11C Fig). Such habitat fragmentation likely contributed to adaptive divergence, however, throughout the entire time series there remain areas between east and west Eurasia where potential suitability was likely sufficient (> 0.4) to maintain gene flow, preventing full speciation.

At 787,000 years before present (MIS19) there is a loss of broad potential suitability in the Tibetan Plateau region, however, western Eurasia, north east China, northern India, Nepal and North and South Korea maintain high potential suitability (S11C Fig). This same patter occurs 130,000 years before present (S11D Fig). At the height of the Last Glacial Maximum, 21,000 years ago the outline of the descending ice sheet is visible in the Northern latitudes, with a loss of potential suitability observed in the more northerly Eurasian Steppe regions

(S11E Fig). This ice age event also coincides with a reduction in high suitability (>0.7) in north east China. Previous work suggested that *Cannabis* populations may have been driven into glacial refugium in the Caucasus Mountain region in Europe and east of the Himalayan foothills in the Hengduan Mountain region [4]. The Hengduan mountains of southwest China is a biodiversity hotspot [14] and had a high potential suitability during this period likely supporting refugial *Cannabis* populations during the Last Glacial Maximum (LGM)[4]. The Himalayan Mountain system also remained suitable; however, it did exhibit a reduction in the more western regions of the Himachal Pradesh (S11E Fig). Between 17,000 – 300 years ago (S11F-K Fig), potential suitability did not change substantially, likely due to the shorter length of the time steps. Despite the relative stability there was an overall expansion in East Asia during the Heinrich-Stadial (S11F Fig) and contraction during the Bolling-Allerod (Fig 1G). This pattern of expansion in East Asia continued after the Younger Dryas Stadial (12,900-11,700) (S11H Fig) and was visible from the Early-Holocene (S11I Fig), Mid-Holocene (8,326 – 4,200) (S11J Fig), the late Holocene (S11K Fig) and into the Anthropocene (S11L Fig). Variable contribution and AUCs for each era can be found in S12 Fig.

Different eras showed different types of range fluctuations, for example East Asia in particular shows large suitability fluctuations during the LGM (S11E Fig) as compared to the Heinrich Stadial (S11F Fig), Bolling-Allerod (S11G Fig) and Younger Dryas Stadial (S11H Fig). High potential suitability is consistently observed in the Eurasian Steppe region and while the range of potential suitability changes throughout the Holocene (S11I-K Fig), it is maintained through all the timepoints examined here (S11A-K Fig). This is similarly the case for north east China and the Himalayan Mountain system (S11A-K Fig). It is therefore possible that *Cannabis* may have had access to a much wider habitat range during this time than previously thought and perhaps during the LGM (S11E Fig). The models here also correspond well to the subfossil pollen records which converge at the northeastern Tibetan Plateau as a proposed center of origin [17]. From here *Cannabis* it is thought to have first dispersed west to Europe by 6 MYA and to eastern China by 1.2 MYA [17]. The development of agricultural practices and the establishment of trade routes along the Eurasian Steppe (e.g., Silk Road) likely facilitated range expansion. Archeological evidence, including carbon dated pollen samples across the species distribution range modeled here is still needed to support and validate the findings presented in this study.

**Current Habitat Suitability of *Cannabis*.** Distributions were constructed using current (1970—2000) bioclimatic (temperature and precipitation) and soil properties separately (S2A-B, S3A-B, S4A-B Fig) and together (Fig 1) to explore global suitability. Six temperature and precipitation variables (BIO-1, BIO-11, BIO-10, BIO-18, BIO-19, BIO-14 - see S4 Table for definitions) explained ~ 81% of the total variation (S3A Fig), with an AUC of 0.9 (S4A Fig). When exploring bioclimatic variables alone, highest suitability was found in mixed deciduous forest, temperate forest steppe and taiga of Eurasia and North America (S2A Fig). When exploring soil alone, four variables (ORCDRC, PHIHOX, OCDENS, CECSOL - see S5 Table for definitions) explained ~ 79% of the total variation (S3B Fig) with an AUC of 0.939 (S4B Fig). Here suitability was also found in the mixed deciduous forest, temperate forest steppe and taiga of Eurasia and North America (Fig 2B).

When climate and soil variables were combined (Fig 1) the most important variables were soil organic carbon content (ORCDRC), pH index measured in water solution (PHIHOX), annual mean temperature (BIO-1), mean temperature of the coldest quarter (BIO-11) and soil organic carbon density (OCDENS) (Fig 1C). With an AUC of 0.934 these five variables explained ~ 65% of the total variation (Fig 1D). Integrating these two datasets we see a notable improvement in model performance (Fig 1A), specifically in mitigating the occurrence of anomalous suitable areas, for example those previously identified at high latitudes of Northern

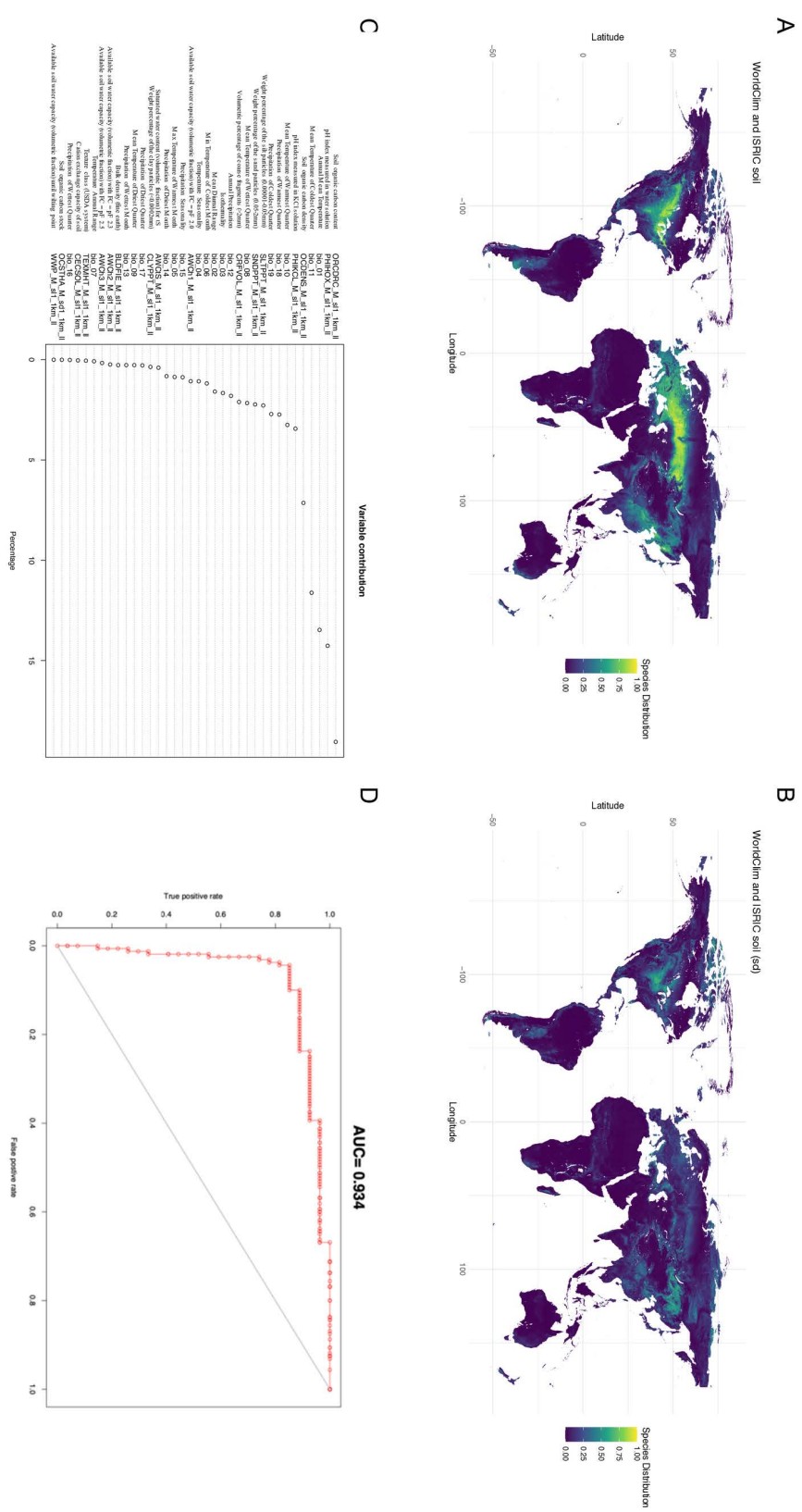

**Fig 1. Maxent generated mean of WorldClim Bioclimatic (temperature and precipitation) variables and ISRIC soil variables for the present day together. (A)** Maxent generated suitability map for *Cannabis* cultivation

worldwide for the present day using WorldClim version 2 climatic and ISRIC soil variable data compiled (**B**) Standard deviation for the merge of the WorldClim and ISRIC SoilGrid variables together. (**C**) Variable contribution of the WorldClim and ISRIC SoilGrid variables together (**D**) AUC for the model.

Canada for the ISRIC soil dataset when examined in isolation (S2B Fig). *Cannabis* is thought to have had a Central Asian Origin [11,33] or an origin region spanning Eurasia [34,35]. Here, high broad suitability (above 0.4) is seen in these regions with a range spanning Europe (S6 Fig), Russia and Asia (Fig 1A, 4A, S6). The patterns of suitability observed matches present-day reports of extant *Cannabis* populations [4]. High suitability is also seen in other continents such as regions of North America (Fig 1A, 4A, S9).

To examine how other environmental variables may affect the model, additional environmental properties namely; solar radiation (kJm -2/day) (S2C Fig), wind speed (m/s) (S2D Fig), water vapor pressure (kPa) (S2E Fig), and elevation (m) (S2F Fig) were acquired. Bringing together all of these environmental properties, a similar pattern of suitability is observed across Eurasia and North America. However, there is a more pronounced variability in the standard deviation for the model worldwide (S5B Fig) in contrast to the more localized variations seen in the bioclimatic and soil standard deviations (Fig 1B). Examining variable contribution across all 73 layers (S6A Fig) shows that 6 of the top 10 contributors are bioclimatic variables (BIO-1, BIO-9 and BIO-11) and soil (ORCDRC, PHIHOX, PHIKCL) properties (Fig 6A) and suggests that integrating bioclimatic and soil properties may reduce background errors that could arise when a broader array of variables is included in the modeling process and which have low variable contributions.

In the important production area of California, good suitability occurs in many counties including Sonoma, Napa, Marin County, San Meteo, Almeda, Monterey, Suter, Yuba and Butte, San Luis Obispo, Santa Barbara, Placer, Mendocino and Humboldt County and almost all coastal counties (Fig 4C, S10A). Additional states which are known for outdoor cultivation were also examined for suitability for all six environmental properties examined namely; Colorado (S10B Fig), Maine (S10C Fig), Oregon (S10D Fig), Washington (S10E Fig ), Massachusetts (S10F Fig) and Michigan (S10G Fig).

**Suitability and Soil.** Like many plant species, making use of soil data increases distribution model performance [36]. Important soil determinants for *Cannabis* suitability included soil pH, soil organic carbon, and cation exchange capacity (CEC) (S3B Fig), which together regulate soil fertility and nutrient cycling in soil. Others have previously reported that *Cannabis* favors growing in alluvial soils under slightly acidic conditions in temperate and subtropical regions [4]. Alluvial soils can be rich in organic matter and plant nutrients, and thus relatively fertile [37] and such soils are found along the Himachal Pradesh [38] and in many places within the global model that have good suitability for *Cannabis* cultivation. Taxonomically, alluvial soils are diverse and fall under an assortment of soil classes [37]. Likewise, in the case of *Cannabis,* higher suitability is observed in regions defined by an array of classifications, including the soil orders of Mollisols (Central Eurasian Steppe and North America), Inceptisols and Ultisols (both found predominantly in southern Russia, Mongolia, and China), with some smaller suitable regions found in areas consisting of Alfisols and Spodosols (Western Eurasian steppe) (S2B Fig). This range in soil taxonomic classifications may highlight the plastic and generalist nature of *Cannabis.* In California, we found a correlation between *Cannabis* suitability and the soil orders of Inceptisols, Mollisols, Alfisols, Aridisols, and Ultisols (Fig 4C-D). The weakly developed Inceptisols of California have a wide range of characteristics, but are mostly associated with steeply sloped chaparral or montane conifer forests and along streams [39]. Certain Inceptisols in Northern California

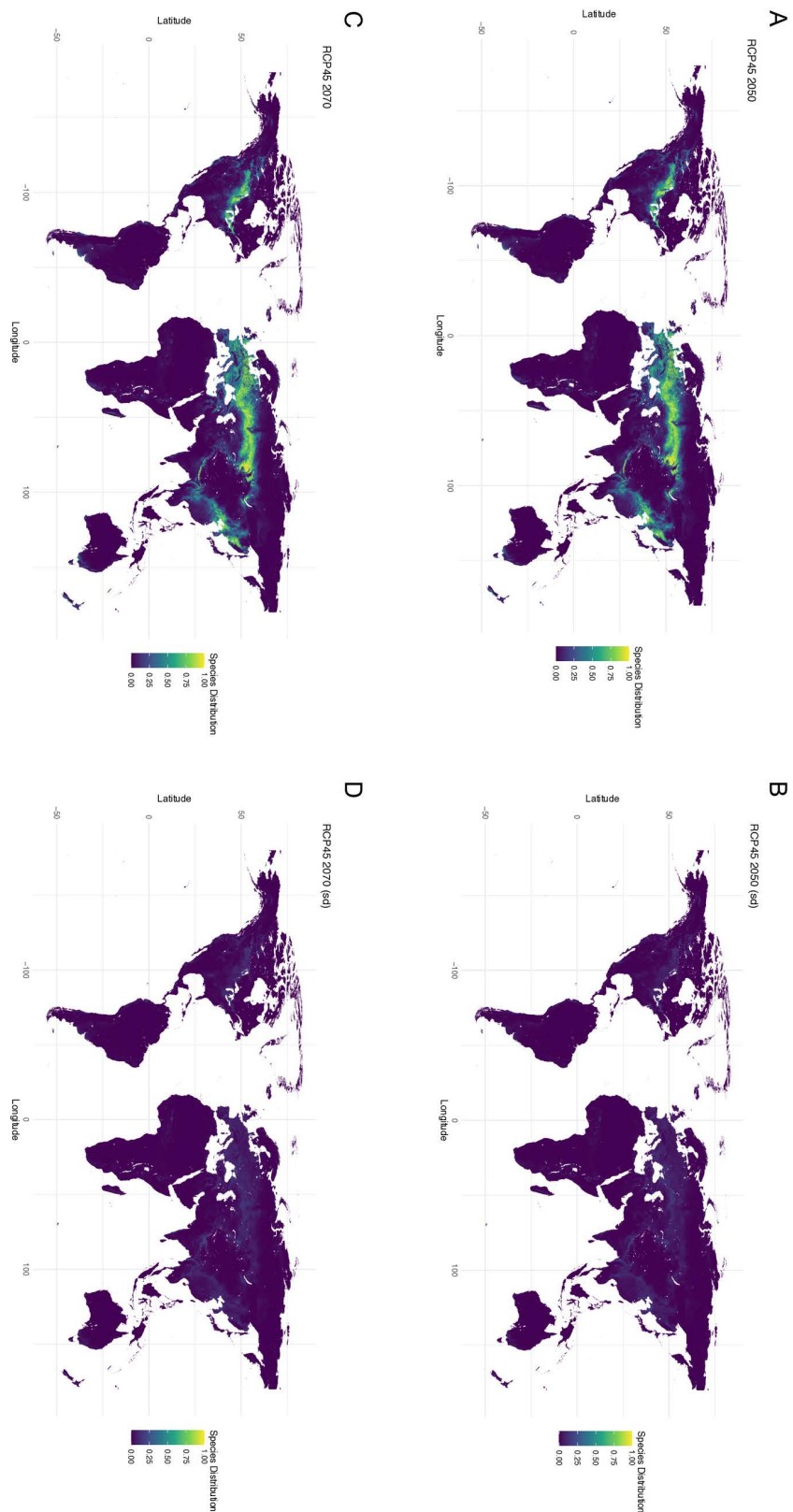

**Fig 2. Maxent generated mean of WorldClim Bioclimatic (temperature and precipitation) variables for future climate projections for 2050 and 2070 with model SSP245 (efforts made to curb climate change). (A)** Worldwide

suitability map for 2050 **(B)** Standard deviation for the worldwide suitability map for 2050 **(C)** Worldwide suitability map for 2070 **(D)** Standard deviation for the worldwide suitability map for 2070.

were also developed from volcanic deposits (Andisols) and have slightly acidic pH with highly variable cation exchange capacity. Mollisols define a large proportion of the suitability score that we see in California (Fig 4C-D), North America (Fig 2B), and the Eurasian Steppe ( S2B Fig). Mollisols are inherently fertile with over 50% base saturation, abundant organic matter, and near neutral pH; whereas moderately weathered Alfisols have more than 35% base saturation with clay rich subsoil horizons with high cation exchange capacity having undergone less intensive leaching than Ultisols [40,41]. There are a number of limitations and biases associated with the climate and soils data that need to be considered. The limitations include the resolution of the bioclimatic and biophysical data and the assumptions made in the interpolation of these data. This is particularly true for biophysical (soil) variables, which likely vary over finer scales than the database which impact the interpolation techniques [42]. Further, the limited sample size of the wild occurrences may lead to increased variability.

**Suitability of *Cannabis under Future Climate Scenarios.*** Future distributions were constructed for both 2050 and 2070 under two emission scenarios (SSP245 - curbing emissions and SSP585 - business as usual). Under SSP245 there is decreasing suitability by 2050 (Fig 2A) and 2070 (Fig 2C). Under SSP585 there is an even greater decrease in suitability by 2050 (Fig 3A) and 2070 (Fig 3B). Future climate datasets represent climatic variables only, as soil properties are more challenging to estimate. Despite the loss of suitability in the future there was overlap between suitability ranges in 2050 and 2070 for both SSP scenarios (Fig 4A).

Partitioning by subregions such as Asia and Russia (S7A-E Fig), Europe and SE Asia (S8 A-E Fig) and the United States (S9A-E Fig) allows for a more quantitative examination of the changes these areas may experience from the present day moving into 2050 and 2070. The rasters generated for each climate scenario (SSP45 and SSP85) and each future time point (2050 & 2070) were filtered for suitability values above 0.4, which is considered an acceptable suitability value for natural growth. Pixel counts were performed and converted into $km^2$, accounting for the curvature of the earth within the range of latitudes in the partition. World-wide across all climate models and timepoints we see an average of ~43% reduction in suitable area (S7 Table) with a reduction from ~13.8 million to ~7.8 million $km^2$. Within the partition of Asia and Russia a ~29% reduction in suitable area from ~6.4 million to ~4.5 million $km^2$ is observed (S7A-E Fig, S7 Table) and similarly in Europe ~15% losses are observed with a reduction from ~2.5 million to ~2.2 million $km^2$ (S8A-E Fig, S7 Table). In the United States an average of ~81% loss in suitable area is observed with a reduction from ~2.8 million to ~0.5 million $km^2$ (S9A-E Fig, S7 Table).

Within the United States, there was a contraction in suitability (Fig 4B, S8A-E Fig), although, suitability remains favorable in areas of California (Fig 4C), with many important production counties showing favorable suitability scores (e.g., Humboldt, Mendocino Sonoma, Marin, Santa Barbara, and Ventura) (Fig 4C). Again, *Cannabis* suitability was correlated with areas where Inceptisols, Mollisols, Vertisols, Ultisols and Alfisols are predominantly found (Fig 4C-D).

**Implications for Future Cultivation.** Globally combining climate and soil variables provides improved resolution (Fig 1A). The observed suitable areas (>0.4) match common agricultural lands known for *Cannabis* cultivation in Asia and Europe and captured specific geographies where genetic studies have shown the presence of drug-type feral *Cannabis* populations [16]. The overlap between suitability for 2050 and 2070 suggests that the effects on *Cannabis* distribution due to changing climate will have mostly taken effect by 2050

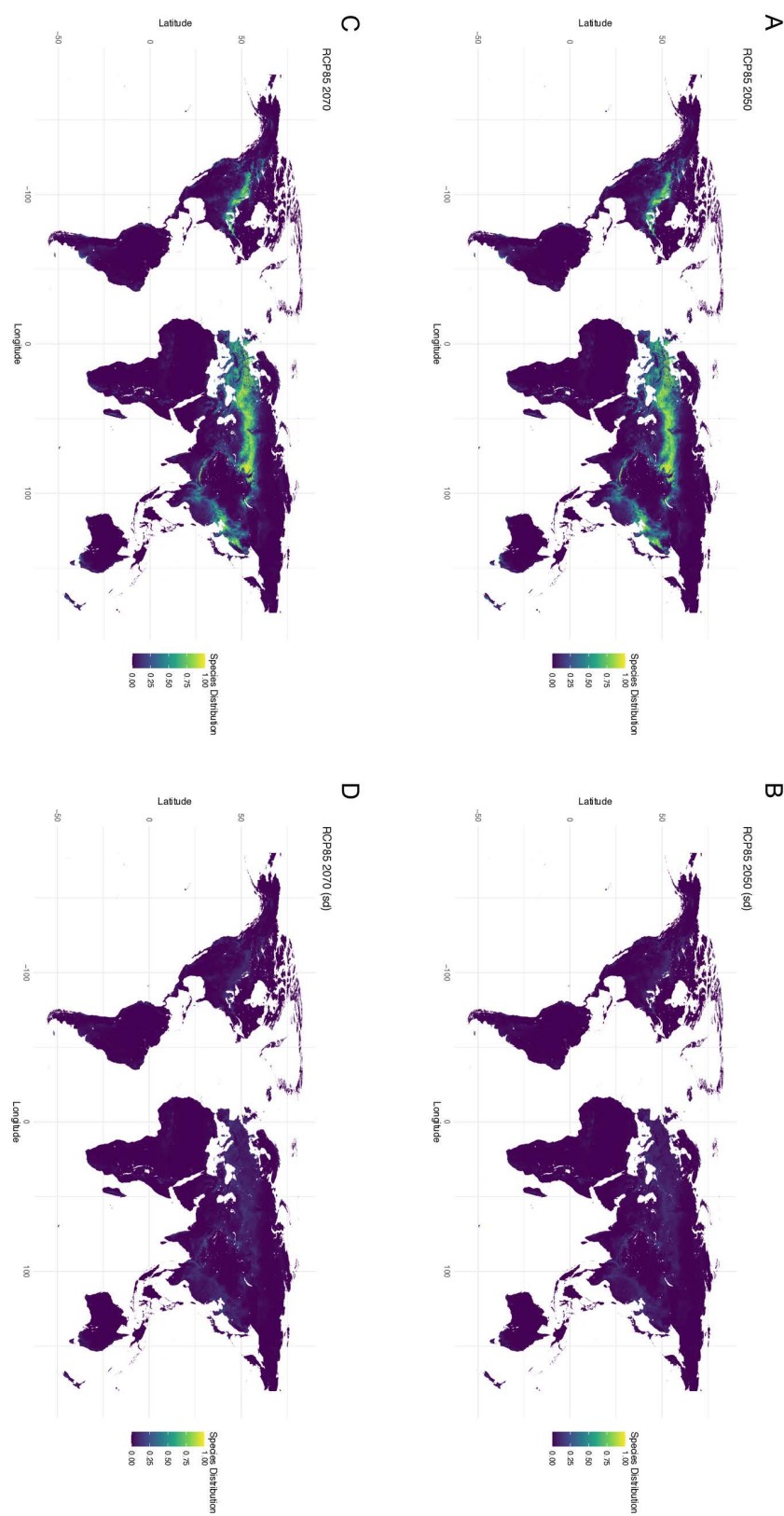

**Fig 3. Maxent generated mean of WorldClim Bioclimatic (temperature and precipitation) variables for 2050 and 2070 with model SSP845 (business-as-usual climate model). (A)** Worldwide suitability map for 2050 **(B)** Standard

deviation for the worldwide suitability map for 2050 **(C)** Worldwide suitability map for 2070 **(D)** Standard deviation for the worldwide suitability map for 2070.

([Fig 4]). Moving into the future, in North America, the states of North Dakota, Minnesota, Wisconsin, and Michigan show high suitability ([Fig 4B]).

Areas in the states of Colorado, Wyoming and California show moderate suitability for the present day, however, moving into the future these areas show decreasing suitability ([S9] Fig). California is known worldwide for its cultivation of *Cannabis*. Proposition 64 in 2016 brought legalization to the recreational use of *Cannabis* and saw a movement towards legal operations in the production and sales of *Cannabis*. Exploring this region more specifically we observed an increase in suitability from the present day to 2050 and 2070 in areas such as Marin County, Contra Costa, San Mateo, Santa Barbara and Ventura ([Fig 4C]). Counties which currently produce *Cannabis* may be impacted under different climate change scenarios in the future. However, future models here only take into account abiotic factors and biotic pressures will also have significant roles in future species distribution.

## Conclusion

Global suitability for *Cannabis* was explored with the intent of identifying regions where *Cannabis* cultivation could be facilitated by suitable climate and soil properties. Across Eurasia and the United States there will be broad loss of suitability by 2050 for two different emission scenarios (SSP245 and SSP585). As suitable habitat decreases, conservation of wild relatives and naturalized populations of *Cannabis* around the world will be critical for the preservation of diversity and act as a valuable source of variation for trait improvement as the industry continues to develop. Similar to previous work we see an expansion and contraction at known times of historical climate fluctuations [43]. Using bioclimatic and biophysical variables has shown to be effective in conservation efforts to help identify areas that should have priority for conservation [44, 45]. This principal has the same potential benefit for identifying priority areas for cultivation under climate change.

## Supplemental information

**S1 Fig.** 137 observations used for SDM model construction with a longitude greater than zero. **Fig S2.** Individual species distributions for each set of environmental properties examined **(A)** WorldClim Bioclimatic variables **(B)** ISRIC soil data **(C)** Solar radiation (kJm$^2$/day) **(D)** Wind speed (m/s) **(E)** Water vapor pressure (kPa) **(F)** Elevation suitability maps. These maps were generated with Maxent using Worldclim and ISRIC data. **Fig S3.** Variable contribution graphs for each set of environmental properties examined **(A)** WorldClim Bioclimatic variables **(B)** ISRIC soil data **(C)** Solar radiation (kJm$^2$/day) **(D)** Wind speed (m/s) **(E)** Water vapor pressure (kPa) and **(F)** Elevation. **Fig S4.** Area under the curve graphs each set of environmental properties examined **(A)** WorldClim Bioclimatic variables **(B)** ISRIC soil data **(C)** Solar radiation (kJm$^2$/day) **(D)** Wind speed (m/s) **(E)** Water vapor pressure (kPa) and **(F)** Elevation. **Fig S5.** Overlay of all six environmental datasets **(A)** Worldwide plot **(B)** standard deviation for the overlay of all six environmental variables. These maps were generated with Maxent using Worldclim and ISRIC data. **Fig S6.** Overlay of all six environmental datasets **(A)** Variable contribution graph **(B)** Area under the curve graphs each set of environmental properties examined. **Fig S7.** Species distribution with temperature and precipitation data in Asia and Russia for **(A)**

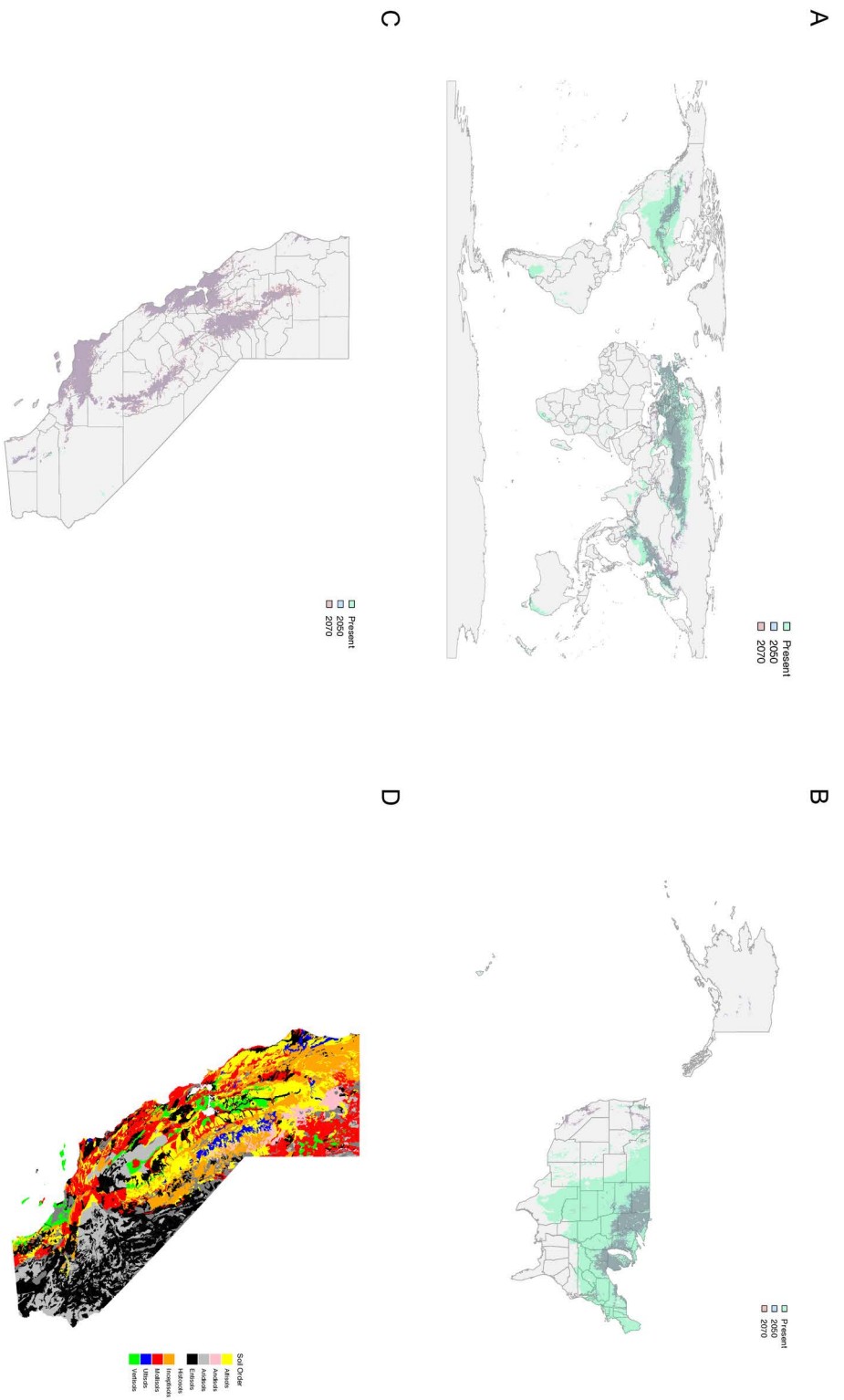

**Fig 4. Suitability overlay with the present day (green) and future projections for *Cannabis sativa* species distribution for 2050 (blue) and 2070 (red) for SSP585 or the business-as-usual climate model.** **(A)** Worldwide suitability map **(B)** Suitability map for the United States and **(C)** Suitability map for the state of California, where green represents present day suitability (above 0.2), Blue represents suitability (above 0.2) for 2050, red suitability

(above 0.2) for 2070 and purple the overlap of suitability for 2050 and 2070 **(D)** Soil orders for the state of California. Maps were generated using Maxent generated using WorldClim Bioclimatic (temperature and precipitation) variables and ISRIC.

present day **(B)** SSP45 2050 **(C)** SSP45 2070 **(D)** SSP85 2050 **(E)** SSP85 2070. These maps were generated with Maxent using Worldclim data. **Fig S8.** Species distribution with temperature and precipitation data in Europe for **(A)** present day **(B)** SSP45 2050 **(C)** SSP45 2070 **(D)** SSP85 2050 **(E)** SSP85 2070. These maps were generated with Maxent using Worldclim data. **Fig S9.** Species distribution with temperature and precipitation data in the United States for **(A)** present day **(B)** SSP45 2050 **(C)** SSP45 2070 **(D)** SSP85 2050 **(E)** SSP85 2070. These maps were generated with Maxent using Worldclim data. **Fig S10.** Species distribution for a subset of the United States with data for all six environmental properties examined **(A)** California **(B)** Colorado **(C)** Maine **(D)** Oregon **(E)** Washington **(F)** Massachusetts **(G)** Michigan. These maps were generated with Maxent using Worldclim data. **Fig S11. (A)** Pleistocene: M2 (ca. 3.3 Ma) **(B)** Predicted Distribution for the Paleoclimate timepoint of the Mid Pliocene warm period (ca. 3.2 Ma) **(C)** Predicted Distribution for the Paleoclimate timepoint of the Pleistocene: MIS19 (ca. 787,000 years ago) **(D)** Predicted Distribution for the Paleoclimate timepoint of the Pleistocene: Last Interglacial (130,000 years ago) **(E)** Predicted Distribution for the Paleoclimate timepoint of the Pleistocene: Last Glacial Maximum (ca. 21,000 years ago) **(F)** Potential Distribution for the Paleoclimate timepoint of the Pleistocene: Heinrich Stadial (14,700 – 17,000 years ago) **(G)** Potential Distribution for the Paleoclimate timepoint of the Pleistocene: Bolling-Allerod (12,900 – 14,700 years ago) **(H)** Potential Distribution for the Paleoclimate timepoint of the Pleistocene: Younger Dryas Stadial (11,700 – 12,900 years ago) **(I)** Potential Distribution for the Paleoclimate timepoint of the Pleistocene: Early Holocene, Greenlandian (8,366 - 11,700 years ago) **(J)** Potential Distribution for the Paleoclimate timepoint of the Pleistocene: Mid Holocene, Northgrippian (4,200 – 8,326 years ago) **(K)** Potential Distribution for the Paleoclimate timepoint of the Pleistocene: Late Holocene, Meghalayan (300 – 4200 years ago). These maps were generated with Maxent using Worldclim data. **Fig S12.** AUC and variable contribution graphs for each timepoint from the Paleoclim dataset **(A)** Pleistocene: M2 (ca. 3.3 Ma) **(B)** Mid Pliocene warm period (ca. 3.2 Ma) **(C)** Pleistocene: MIS19 (ca. 787,000 years ago) **(D)** Pleistocene: Last Interglacial (130,000 years ago) **(E)** Last Glacial Maximum (ca. 21,000 years ago) **(F)** Heinrich Stadial (14,700 – 17,000 years ago **(G)** Bolling-Allerod (12,900 – 14,700 years ago) **(H)** Younger Dryas Stadial (11,700 – 12,900 years ago) **(I)** Early Holocene, Greenlandian (11,700-8,326 years ago) **(J)** Mid Holocene, Northgrippian (4,200 – 8,326 years ago) **(K)** Late Holocene, Meghalayan (300 – 4200 years ago).
(ZIP)

**S2 File.** **Table** S1. 416 occurrence points from iNaturalist which had paired images. **Table S2.** Latitude and longitudes of 302 occurrence points deemed to be growing wild without human intervention. **Table S3.** Latitude and Longitude for the 137 observations used for SDM model construction, post-filtering for a longitude greater than zero. **Table S4.** WorldClim2 Bioclimatic variable definitions. **Table S5.** ISRIC soil variable definitions. **Table S6.** Paleoclimate Time Period and date range **Table S7.** Pixel counts for present day, SSP 45 and SSP85 for 2050 and 2070 above the 0.4 threshold for the world and the partitions of Asia & Russia, Europe and SE Asia and the United States
(XLSX)

## Acknowledgments

We would like to thank Koa the University of Hawai'i (UH) high performance computing (HPC) cluster.

## Author contributions

**Conceptualization:** Anna Halpin-McCormick, Tai McClellan Maaz, Michael B. Kantar, Rishi R. Masalia, Nick Batora, Kerin Law, Eleanor J. Kuntz.

**Data curation:** Anna Halpin-McCormick, Tai McClellan Maaz, Michael B. Kantar.

**Formal analysis:** Anna Halpin-McCormick, Tai McClellan Maaz.

**Supervision:** Kasey E. Barton, Rishi R. Masalia, Kerin Law, Eleanor J. Kuntz.

**Visualization:** Anna Halpin-McCormick.

**Writing – original draft:** Anna Halpin-McCormick.

**Writing – review & editing:** Anna Halpin-McCormick, Tai McClellan Maaz, Michael B. Kantar, Kasey E. Barton, Rishi R. Masalia, Nick Batora, Kerin Law, Eleanor J. Kuntz.

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
