## [Decision Letter · Decision Letter 0]

1 Aug 2024

PONE-D-24-23287Species Distribution of Cannabis sativa: Past, Present and futurePLOS ONE

Dear Dr. Kantar,

Thank you for submitting your manuscript to PLOS ONE. After careful consideration, we feel that it has merit but does not fully meet PLOS ONE’s publication criteria as it currently stands. Therefore, we invite you to submit a revised version of the manuscript that addresses the points raised during the review process.

We look forward to receiving your revised manuscript.

Kind regards,

Andrea Mastinu

Academic Editor

PLOS ONE

Journal Requirements:

2. Thank you for stating the following in the Competing Interests section: "LeafWorks Inc. is a for profit company."

We note that you received funding from a commercial source: LeafWorks Inc.

Within this Competing Interests Statement, please confirm that this does not alter your adherence to all PLOS ONE policies on sharing data and materials by including the following statement: ""This does not alter our adherence to PLOS ONE policies on sharing data and materials.” (as detailed online in our guide for authors http://journals.plos.org/plosone/s/competing-interests).  If there are restrictions on sharing of data and/or materials, please state these. Please note that we cannot proceed with consideration of your article until this information has been declared. 

4. We note that Figures 1,2,3,4, S1,S2,S5,S7,S8,S9,S10 and S11 in your submission contain [map/satellite] images which may be copyrighted. All PLOS content is published under the Creative Commons Attribution License (CC BY 4.0), which means that the manuscript, images, and Supporting Information files will be freely available online, and any third party is permitted to access, download, copy, distribute, and use these materials in any way, even commercially, with proper attribution. For these reasons, we cannot publish previously copyrighted maps or satellite images created using proprietary data, such as Google software (Google Maps, Street View, and Earth). For more information, see our copyright guidelines: http://journals.plos.org/plosone/s/licenses-and-copyright.

1. You may seek permission from the original copyright holder of Figures 1,2,3,4, S1,S2,S5,S7,S8,S9,S10 and S11 to publish the content specifically under the CC BY 4.0 license.  

Reviewers' comments:

Reviewer's Responses to Questions

**Comments to the Author**

1. Is the manuscript technically sound, and do the data support the conclusions?

Reviewer #1: Yes

2. Has the statistical analysis been performed appropriately and rigorously? 

Reviewer #1: I Don't Know

3. Have the authors made all data underlying the findings in their manuscript fully available?

Reviewer #1: Yes

4. Is the manuscript presented in an intelligible fashion and written in standard English?

Reviewer #1: Yes

5. Review Comments to the Author

Reviewer #1: The article attempts to depict the past, present, and future distribution of hemp species using available global data. Although the paper is well-structured, I found the results and discussion sections difficult to follow due to the numerous figures and tables. I suggest adding a separate section focused directly on the discussion. The methodology is straightforward, but there is a significant gap in training data from other continents, which the authors did not address or discuss the implications of. While the authors used freely available global data, the accuracy of this data, especially soil information, is questionable and should be properly discussed in the paper.

Introduction

- "Cannabis now considered a monotypic genus". This is still a matter of debate, particularly when we consider Cannabis ruderalis. Please add more context here.

- "Identifying the best climatic and soil conditions for growth informs nutrient and water management and approaches such as species distribution modeling (SDM) providing a means for more informed land selection that may facilitate minimizing negative environmental externalities (Mehrabi et al., 2019)" This sentence contaisn many ideas and the reference seems to be for sunflower. Please re-write it and provide a more relavant reference.

- Instead of just pasting the link, provide a proper citation for the Economist article about Cannabis market in California.

- Whilst the introduction section is well organised, it lacks depth and proper linkage to the 'knowledge gap' the article is trying to fill. More importantly, the focus of the article seems to be a study of status and shift of cultivation at the global level. Bud the authors suddenly add information about the cultivation center in California. I suggest the authors expand the introduction section and provide a better linkage to the aim and objectives of the study.

Methodology

- Please provide a proper citation for the GBIF occurrence data.

- Were there other occurrence data for C. Sative on GBIF?

- "Of these, 302 were deemed as wild or escapees and growing without human intervention " how they were deemed wild? what was the method?

- Table Table S3 137 in Supplementary Materials is empty!

- The ISRIC soil data were old. Can you use OpenLandMap or https://soil.copernicus.org/articles/7/217/2021/?

- Please create proper citations for URLs in the methodology section.

- Why different sources was used for admin boundaries shapefiles? Was GADM database not comprehensive enough?

- Please add a better description of how historical SDMs were created. How they were validated?

Results and discussion

- Although both "suitability" and "species occurrrence" have been used interchangeably, suitability encompasses a much bigger area and in this context, I advice authors to avoid use of the word 'suitablity', particularly when they are referring to the past coditions.

Other comments:

- References are not ordered. That makes it difficult for the people to find

- There is no record in bibliography linked to Clare and Merlin, 2013: Do you mean Clarke, R. C., & Merlin?

6. PLOS authors have the option to publish the peer review history of their article (what does this mean? ). If published, this will include your full peer review and any attached files.

**Do you want your identity to be public for this peer review?** For information about this choice, including consent withdrawal, please see our Privacy Policy .

Reviewer #1: No

---

## [Author Response · Author response to Decision Letter 0]

20 Aug 2024

Response To reviewer Comments

Editor:

Thank you for the opportunity to revise the manuscript. The comments were very helpful and we feel they have greatly improved the manuscript.

Comment 1: When submitting your revision, we need you to address these additional requirements. Please ensure that your manuscript meets PLOS ONE's style requirements, including those for file naming. The PLOS ONE style templates can be found at

Response: We have modified the manuscript to fit the template and the reference scheme changed to numbers

Comment 2: Thank you for stating the following in the Competing Interests section: "LeafWorks Inc. is a for profit company." We note that you received funding from a commercial source: LeafWorks Inc. Please provide an amended Competing Interests Statement that explicitly states this commercial funder, along with any other relevant declarations relating to employment, consultancy, patents, products in development, marketed products, etc. Within this Competing Interests Statement, please confirm that this does not alter your adherence to all PLOS ONE policies on sharing data and materials by including the following statement: ""This does not alter our adherence to PLOS ONE policies on sharing data and materials.” (as detailed online in our guide for authors http://journals.plos.org/plosone/s/competing-interests). If there are restrictions on sharing of data and/or materials, please state these. Please note that we cannot proceed with consideration of your article until this information has been declared. Please include your amended Competing Interests Statement within your cover letter. We will change the online submission form on your behalf.

Response: We have changed the Conflict of Interest Statement to

“LeafWorks is a for profit company that provided funding to help with the computing for this manuscript. This does not alter our adherence to PLOS ONE policies on sharing data and materials.”

Comment 3: We note that Figures 1,2,3,4, S1,S2,S5,S7,S8,S9,S10 and S11 in your submission contain [map/satellite] images which may be copyrighted.

Response: These images do not actually contain satellite data, the input rasters are from worldclim and ISRIC and the images and colouring generated is based on the analysis done in this manuscript using Maxent. The viridis palette was used in the final plots.

Comment 4: Please include captions for your Supporting Information files at the end of your manuscript, and update any in-text citations to match accordingly. Please see our Supporting Information guidelines for more information: http://journals.plos.org/plosone/s/supporting-information.

Response: We have added the captions.

Reviewer #1 comments:

Comment 1: The article attempts to depict the past, present, and future distribution of hemp species using available global data. Although the paper is well-structured, I found the results and discussion sections difficult to follow due to the numerous figures and tables. I suggest adding a separate section focused directly on the discussion. The methodology is straightforward, but there is a significant gap in training data from other continents, which the authors did not address or discuss the implications of. While the authors used freely available global data, the accuracy of this data, especially soil information, is questionable and should be properly discussed in the paper.

Response: Thank you for this summary. We have added cavates regarding the soil data. The below has been added to the end of the suitability and soil section in the results & discussion section for this purpose.

“There are a number of limitations and biases associated with the climate and soils data that need to be considered. The limitations include the resolution of the bioclimatic and biophysical data and the assumptions made in the interpolation of these data. This is particularly true for biophysical (soil) variables, which likely vary over finer scales than the database which impact the interpolation techniques (Brady et al. 2005). “

Brady, K. U., A. R. Kruckeberg, and H. D. Bradshaw, Jr., 2005 Evolutionary ecology of plant adaptation to serpentine soils. Annu. Rev. Ecol. Evol. Syst. 36: 243–266.

As regards the significant gap in training data from other continents - as we are dealing with suspected wild plants, we wouldn't expect to use data points from other continents for training as the native range for cannabis is Eurasia. We have added the below into the text to elaborate on why eurasian data points were selected in the materials and methods

“After filtering for a longitude greater than zero, 137 observations remained and were used for SDM model construction (Table S3; Fig. S1). Eurasia is considered the center of origin of Cannabis, which provided rationale for including only data points with a longitude greater than zero in this study.”

Comment 2: "Cannabis now considered a monotypic genus". This is still a matter of debate, particularly when we consider Cannabis ruderalis. Please add more context here.

Response: Cannabis ruderalis could be considered in some more northern Russian latitudes where it grows freely, a wild relative. These plants have been used widely for the introgression of day neutrality in flowering time. Due to this fertility between all cannabis types when crossed (hemp, ruderal, recreational) it is clear there is no species barrier. There is considerable debate about if species delineations are therefore appropriate given this cross variety fertility. Only one paper to date (Ren et al, 2021) includes a genetic comparison of wild or feral, hemp type and THC-dominant type cannabis samples - and while the feral sampled varieties are most basal in the phylogeny, this genetic distance does not seem to affect crossing compatibility. The below has been included in the introduction to add more nuance

“Historically, it has been broadly divided into two types, hemp- or drug-type, with early descriptions dating back to Linnaeus (1753) and Lamarck (1785) (Linnaeus, 1753; Lamark, 1783). Linnaeus described the plants from Northern Europe as Cannabis sativa and Lamarck described plants from India as Cannabis indica. In 1924 an additional purported wild species growing in central Russia was described by Janischewsky (Janischewsky, 1924) and termed Cannabis ruderalis. At the time, morphological differences between these three taxa led to the proposition of multiple species (sativa, indica and ruderalis) however, more recent work supports the rank of subspecies (McParland, 2018) with genetic diversity occurring across a latitudinal gradient along which classic differentiating phenotypes occur (Zhang et al., 2018). Due to the cross fertility of the proposed multispecies, Cannabis is now more commonly considered a monotypic genus (Small, 2017; Barcaccia et al., 2020; Kocalchuck et al., 2020)”

Comment 3: "Identifying the best climatic and soil conditions for growth informs nutrient and water management and approaches such as species distribution modeling (SDM) providing a means for more informed land selection that may facilitate minimizing negative environmental externalities (Mehrabi et al., 2019)"

This sentence contains many ideas and the reference seems to be for sunflower. Please re-write it and provide a more relavant reference.

Response: We have rewritten the sentence

“Under climate change it is very possible that in the future different regions may be more appropriate for different crops (Pironon et al., 2019). Species distribution modeling (SDM) provides a method for informed land selection by identifying regions with favorable climatic and soil conditions and providing a suitability score for identifying these regions.”

Pironon, S., Etherington, T. R., Borrell, J. S., Kühn, N., Macias-Fauria, M., Ondo, I., et al. (2019). Potential adaptive strategies for 29 sub-Saharan crops under future climate change. Nature Climate Change, 9(10), 758-763.

Comment 4: Instead of just pasting the link, provide a proper citation for the Economist article about Cannabis market in California.

Response: We have changed this reference to (Walsh, 2023). Full reference is: Walsh, Dustin. "Michigan's weed market is now the top in the nation." Crain's Detroit Business, vol. 39, no. 33, 28 Aug. 2023, p. 0005. Gale OneFile: Health and Medicine, link.gale.com/apps/doc/A762882796/HRCA?u=albe12389&sid=googleScholar&xid=f17ee90f.”

No issue number could be found for the economist article or author, only the date it was published online: May 14th 2022 (https://www.economist.com/united-states/2022/05/14/in-california-the-worlds-largest-legal-weed-market-is-going-up-in-smoke)

Comment 5: Whilst the introduction section is well organised, it lacks depth and proper linkage to the 'knowledge gap' the article is trying to fill. More importantly, the focus of the article seems to be a study of status and shift of cultivation at the global level. Bud the authors suddenly add information about the cultivation center in California. I suggest the authors expand the introduction section and provide a better linkage to the aim and objectives of the study.

Response: The knowledge gap and rationale for this study was that no prior species distribution modeling for cannabis has been published for present day or future climate scenarios. We aimed to provide a quantitative analysis on how favorable ranges of Cannabis may change moving towards the various future climate scenarios. The introduction has been amended to include the below to better link the aim and objectives of the study and why the focus was directed to California more specifically

“Despite this widespread cultivation, no prior species distribution modeling for Cannabis has been published for present day or future scenarios”

“Species distribution modeling (SDM) provides a method for informed land selection by identifying regions with favorable climatic and soil conditions and providing a suitability score for identifying these regions. SDM can also support conservation decision making by identifying where other suitable habitats may be for endangered species and identifying the rank in important environmental factors involved in their distribution.”

“However it is currently unknown how future climate change may impact Cannabis cultivation and conservation globally and what are the most important environmental variables that explain Cannabis species distribution.”

Comment 6: Please provide a proper citation for the GBIF occurrence data.

Response: GBIF suggests this is how data to be cited: GBIF.org (20 April 2020) GBIF Occurrence Download https://doi.org/10.15468/dl.d8n6hx and this is what is referenced in the document. Please see below image for rationale.

Comment 7: Were there other occurrence data for C. Sative on GBIF?

Response: There are many other occurrence data for C.sativa on GBIF, however only data points which had associated images were included in this analysis, reducing the total number of GBIF records dramatically. Without being able to see the plants surroundings, it would be impossible to know if the plant was growing wild and without human intervention and so occurrence points with no associated image were not included.

Comment 8: "Of these, 302 were deemed as wild or escapees and growing without human intervention " how they were deemed wild? what was the method?

Response: For the data points used in the study, associated images for each latitude and longitude were individually examined and the data point was only included where it was clear the plant was growing wild and most likely without any human intervention. The call on wild was made if no man made objects of any kind were seen in the photo and that the plant appeared to be growing along with others or in a landscape that appeared to not be being managed. In table S1 the associated links to the images for each datapoint that was assessed are included. In this version of the tables, the link to the corresponding images is included for the 137 datapoints (Table S3) used in the downstream analyses.

The following has been added to the materials and methods “Plants were classified as wild if image inspection revealed no visible man-made objects of any kind, the plant was growing amongst other plants and the landscape appeared unmanaged.”

Comment 9: Table Table S3 137 in Supplementary Materials is empty!

Response: Apologies these data points have been included in this version of the tables.

Comment 10: The ISRIC soil data were old. Can you use OpenLandMap or https://soil.copernicus.org/articles/7/217/2021/?

Response: We thank the reviewer for pointing this out. We are not able to redo the analysis with this data, but we have added caveats about the data being used. The following has been added to the text of the discussion.

“There are a number of limitations and biases associated with the climate and soils data that need to be considered. The limitations include the resolution of the bioclimatic and biophysical data and the assumptions made in the interpolation of these data. This is particularly true for biophysical (soil) variables, which likely vary over finer scales than the database which impact the interpolation techniques (Brady et al. 2005).”

Comment 11: Please create proper citations for URLs in the methodology section.

Response: The links to worldclim and isric have been removed leaving just the primary references for these. A citation was possible for the EarthWorks shapefiles, but none was found for the world administrative boundaries shape file, however the link was removed as requested. Remaining links are for the source code github and high resolution figures.

Comment 12: Why different sources was used for admin boundaries shapefiles? Was GADM database not comprehensive enough?

Response: There was no specific reason for using different shapefiles other than prior knowledge of the world administrative boundaries and earthworks Standford sources.

Comment 13: Please add a better description of how historical SDMs were created. How they were validated?

Response: Historical SDM were created similarly to the present day SDM using Maxent. Present day latitude and longitude datapoints (n=137) were used under the assumption that where cannabis grows wild today may reflect where it could have grown in the past. The Paleoclimate was instead the input raster for the conditions at each time point. Without samples with geo-references and pollen dating, it is not possible to validate the historical SDM. This is why they are a supplemental figure. It is acknowledged that assumptions were made about present day and past ranges to conduct this analysis.

The following has been added to the text of the manuscript:

“Paleoclimate data were sourced from paleoclim.org with the highest spatial resolution of 2.5 arc-minutes (~5km) downloaded (Table S6, (Fordham et al., 2917)) with suitability maps were created using the Maxent software (Version 3.3.4). For historic distribution models, present day latitude and longitude data points (n=137) were used under the assumption that where cannabis grows wild today may reflect where it could have grown in the past.”

As well as the following to results and discussion section:

“Archeological evidence, including carbon dated pollen samples across the species distribution range modeled here is still needed to support and validate the findings present

---

## [Decision Letter · Decision Letter 1]

6 Nov 2024

PONE-D-24-23287R1Species Distribution of  Cannabis sativa : Past, Present and futurePLOS ONE

Dear Dr. Kantar,

Thank you for submitting your manuscript to PLOS ONE. After careful consideration, we feel that it has merit but does not fully meet PLOS ONE’s publication criteria as it currently stands. Therefore, we invite you to submit a revised version of the manuscript that addresses the points raised during the review process.

We look forward to receiving your revised manuscript.

Kind regards,

Andrea Mastinu

Academic Editor

PLOS ONE

Journal Requirements:

**Comments from PLOS Editorial Office:**
*We note that one or more reviewers has recommended that you cite specific previously published works. As always, we recommend that you please review and evaluate the requested works to determine whether they are relevant and should be cited. It is not a requirement to cite these works. We appreciate your attention to this request.*

Reviewers' comments:

Reviewer's Responses to Questions

**Comments to the Author**

1. If the authors have adequately addressed your comments raised in a previous round of review and you feel that this manuscript is now acceptable for publication, you may indicate that here to bypass the “Comments to the Author” section, enter your conflict of interest statement in the “Confidential to Editor” section, and submit your "Accept" recommendation.

Reviewer #2: (No Response)

Reviewer #3: (No Response)

Reviewer #4: (No Response)

Reviewer #5: All comments have been addressed

2. Is the manuscript technically sound, and do the data support the conclusions?

Reviewer #2: Yes

Reviewer #3: Yes

Reviewer #4: Yes

Reviewer #5: Yes

3. Has the statistical analysis been performed appropriately and rigorously? 

Reviewer #2: Yes

Reviewer #3: Yes

Reviewer #4: Yes

Reviewer #5: No

4. Have the authors made all data underlying the findings in their manuscript fully available?

Reviewer #2: Yes

Reviewer #3: Yes

Reviewer #4: Yes

Reviewer #5: Yes

5. Is the manuscript presented in an intelligible fashion and written in standard English?

Reviewer #2: Yes

Reviewer #3: Yes

Reviewer #4: Yes

Reviewer #5: Yes

6. Review Comments to the Author

Reviewer #2: The manuscript entitled “Species Distribution of Cannabis sativa: Past, Present and future" used species distribution modeling approach together with relevant climate variables to construct the past and predict the current and future distribution patterns of Cannabis sativa. The manuscript is well written, and the drawn conclusions are coherent with the obtained results. Although similar methodologies are common, the results of the study could have useful implications for management actions. The manuscript requires some changes before it's ready for publication.

Abstract:

I suggest reporting the habitat areas (changes) in Kilometers for the past, current, and the future.

Introduction

- “(12)(Zhang et al., 2018).” Please either use ‘numbered’ reference style or ‘authors, date’ style.

Material and Methods

- “species distribution models (SDM)” should be “species distribution models (SDMs)”

- I suggest restructuring the methodology section into the following subsection:

1- Occurrence points

2- Environmental variables

3- Model building

4- Model evaluation (Area Under the Receiver Operating Curve (AUC)).

In this section, it's important to clarify what threshold was used to delineate the suitability and unsuitability areas.

Discussion

Implications for Future Cultivation : A small section highlighting the benefits of the applied modeling techniques in establishing priority zones for management actions is necessary. In addition, a couple of sentences on the limitations of the modeling technique is required. For this, I suggest:

https://doi.org/10.1016/j.ecoinf.2022.101930

https://doi.org/10.1007/s10661-024-12438-z

https://doi.org/10.1007/s10661-024-12438-z

https://www.mdpi.com/2071-1050/14/21/14621

Reviewer #3: Species Distribution of Cannabis sativa: Past, Present, and Future (PONE-D-24-23287R1)

This manuscript presents a novel exploration of the historical, current, and projected future distributions of Cannabis sativa. By employing species distribution modeling (SDM) based on bioclimatic and soil variables, it offers insights into how changing climate conditions could impact the range and suitability of Cannabis habitats. The modeling under paleoclimatic scenarios, coupled with present and future climate projections, adds significant value to understanding both the ecological and practical dimensions of Cannabis cultivation and conservation.

The revised manuscript addressed many initial comments, particularly in methodology justification, result interpretation, and improvements to data selection rationale. The addition of comments on data limitations, particularly regarding soil information and gaps in training data, enhances the transparency of the methodology. However, some amendments remain necessary for clarity and accuracy.

I recommend that this revised manuscript be accepted for publication in PLOS ONE after addressing some remaining amendments for clarity and scientific rigor. The authors have responded well to previous comments, but additional refinements would further strengthen the manuscript.

• The manuscript currently includes both numerical citations and author-based in-text citations (e.g., “Zhang et al., 2018”), which creates inconsistency. Please unify the reference style, choosing either numerical or author-date citation for all references in accordance with PLOS ONE formatting guidelines.

• The manuscript would benefit from a clearer distinction between Cannabis species growing in the wild and those cultivated under controlled conditions. This differentiation is crucial, as cultivation practices allow for precise management of variables, whereas wild populations are subject to natural selection pressures and environmental variability. Expanding this discussion could clarify the ecological impacts and distinctions in Cannabis adaptation strategies.

• The authors used 137 occurrence points to model habitat suitability globally, which may be a limited representation, especially given the scale of the analysis. A discussion addressing whether this sample size sufficiently captures global suitability would be valuable, perhaps mentioning potential limitations and the implications of using Eurasian data exclusively for a global SDM.

• Although the paper explores past and present distribution shifts, it lacks a focused discussion on historical changes in the distribution of Cannabis. Integrating an analysis or discussion of historical fluctuations in its range, especially in response to past climatic shifts, could provide a more comprehensive context.

• While the authors effectively discuss how climate variables have influenced Cannabis distribution, soil properties such as organic carbon content and pH also vary over millennia. A brief discussion on how soil characteristics have changed historically could help contextualize the model’s results, as climate and soil variability are interrelated in shaping Cannabis habitats.

Reviewer #4: The manuscript, titled "Species Distribution of Cannabis sativa: Past, Present and Future," offers valuable insights into how this species’ distribution responds to climate change. By modeling the historical, current, and projected future ranges of Cannabis sativa, the study highlights key environmental factors that shape its distribution under different climate scenarios. This research adds meaningfully to our understanding of the expansion-contraction model, especially for species affected by environmental shifts.

In terms of modeling, the use of AUC as a performance measure is helpful; however, including AICc and ROC values would make the model evaluation even more robust. AICc would allow for a clearer comparison of model fit across alternatives, while the ROC curve would provide a more complete view of predictive performance. These additions could be especially useful for readers with an interest in model selection and accuracy.

Additionally, the discussion on expansion and contraction might benefit from referencing the study by Ülker, Tavşanoğlu, and Perktaş (2018) on Quercus robur ("Ecological niche modeling of Pedunculate Oak supports the ‘Expansion-Contraction’ model of Pleistocene biogeography"). Including this work may offer a useful comparison and add depth to the discussion of species distribution changes over time.

Finally, it’s clear that the authors have made considerable improvements based on previous feedback, especially in addressing the limitations of soil data, explaining data selection, and refining methodological details. These thoughtful updates have successfully resolved earlier concerns and have strengthened the manuscript overall.

Reviewer #5: In this study, authors have used various multi-source spatial and non-spatial data to investigate species Distribution of Cannabis sativa for Past, Present and future. It is an impressive compilation of information related with the documentation of Cannabis sativa. However, I suggest authors to include/correct following details in the revised manuscript.

1. Authors have used CMIP5 GCM dataset for future period. Cannabis are sensitive to temperature and humidity. Previous CMIP5 models have higher uncertainty in temperature and rainfall than CMIP6 GCMs. I suggest to apply dataset of CMIP6 GCMs, otherwise provide uncertainty in estimated suitability areas due to the use of CMIP5 GCMs.

7. PLOS authors have the option to publish the peer review history of their article (what does this mean? ). If published, this will include your full peer review and any attached files.

**Do you want your identity to be public for this peer review?** For information about this choice, including consent withdrawal, please see our Privacy Policy .

Reviewer #2: No

Reviewer #3: **Yes: ** Hossein Bashari

Reviewer #4: No

Reviewer #5: No

---

## [Author Response · Author response to Decision Letter 1]

6 Nov 2024

Editor:

Thank you for the opportunity to revise the manuscript

Reviewer 2

Comment: The manuscript entitled “Species Distribution of Cannabis sativa: Past, Present and future" used species distribution modeling approach together with relevant climate variables to construct the past and predict the current and future distribution patterns of Cannabis sativa. The manuscript is well written, and the drawn conclusions are coherent with the obtained results. Although similar methodologies are common, the results of the study could have useful implications for management actions. The manuscript requires some changes before it's ready for publication.

Response: Thank you for the succinct summary

Comment: I suggest reporting the habitat areas (changes) in Kilometers for the past, current, and the future.

Response: For the present and day and future scenarios (both rcp and 2050/2070) we have in Table S7 the conversions to km2 worldwide as well as for the niches, as well as for specific areas of interest.

We have not done this for the past data however, as we feel that these estimations in the past for species distribution may not so easily translate accurately, given the changes in geographic landscapes in a deeper time context (rising ocean levels, glacial maximums etc). It may not be a fair estimate to compare the value in range changes in the past to the present given the occurrence of such major geological events outside of just climate properties.

Comment: “(12)(Zhang et al., 2018).” Please either use ‘numbered’ reference style or ‘authors, date’ style.

Response: Done

Comment: “species distribution models (SDM)” should be “species distribution models (SDMs)”

Response: Done

Comment: I suggest restructuring the methodology section into the following subsection:

1- Occurrence points

2- Environmental variables

3- Model building

4- Model evaluation (Area Under the Receiver Operating Curve (AUC)).

Response: Done

Comment: In this section, it's important to clarify what threshold was used to delineate the suitability and unsuitability areas.

Response: Done, as the reviewer probably accidentally overlooked this was already stated

“Suitability maps were overlaid for the present day (1970-2000), 2050 and 2070, with a suitability cutoff score of 0.2. Acceptable suitability is defined as 0.2 for cultivated regions (30) and 0.4 for natural areas (31).”

Comment: Implications for Future Cultivation : A small section highlighting the benefits of the applied modeling techniques in establishing priority zones for management actions is necessary. In addition, a couple of sentences on the limitations of the modeling technique is required. For this, I suggest:

https://doi.org/10.1016/j.ecoinf.2022.101930

https://doi.org/10.1007/s10661-024-12438-z

https://doi.org/10.1007/s10661-024-12438-z

https://www.mdpi.com/2071-1050/14/21/14621

Response: We have added the sentences:

“Using bioclimatic and biophysical variables has shown to be effective in conservation efforts to help identify areas that should have priority for conservation (Hama and Khwarahm, 2023; Mirhashemi et al., 2024). The principal has the same potential benefit for identifying priority areas for cultivation under climate change.”

Hama, A. A., & Khwarahm, N. R. (2023). Predictive mapping of two endemic oak tree species under climate change scenarios in a semiarid region: range overlap and implications for conservation. Ecological Informatics, 73, 101930.

Mirhashemi, H., Ahmadi, K., Heydari, M., Karami, O., Valkó, O., & Khwarahm, N. R. (2024). Climatic variables are more effective on the spatial distribution of oak forests than land use change across their historical range. Environmental Monitoring and Assessment, 196(3), 289.

Reviewer 3

Comment: This manuscript presents a novel exploration of the historical, current, and projected future distributions of Cannabis sativa. By employing species distribution modeling (SDM) based on bioclimatic and soil variables, it offers insights into how changing climate conditions could impact the range and suitability of Cannabis habitats. The modeling under paleoclimatic scenarios, coupled with present and future climate projections, adds significant value to understanding both the ecological and practical dimensions of Cannabis cultivation and conservation.

Response: Thank you for the summary

Comment: The revised manuscript addressed many initial comments, particularly in methodology justification, result interpretation, and improvements to data selection rationale. The addition of comments on data limitations, particularly regarding soil information and gaps in training data, enhances the transparency of the methodology.

Response: Thank you

Comment: I recommend that this revised manuscript be accepted for publication in PLOS ONE after addressing some remaining amendments for clarity and scientific rigor. The authors have responded well to previous comments, but additional refinements would further strengthen the manuscript.

Response: Thank you

Comment: The manuscript currently includes both numerical citations and author-based in-text citations (e.g., “Zhang et al., 2018”), which creates inconsistency. Please unify the reference style, choosing either numerical or author-date citation for all references in accordance with PLOS ONE formatting guidelines.

Response: Done

Comment: The manuscript would benefit from a clearer distinction between Cannabis species growing in the wild and those cultivated under controlled conditions. This differentiation is crucial, as cultivation practices allow for precise management of variables, whereas wild populations are subject to natural selection pressures and environmental variability. Expanding this discussion could clarify the ecological impacts and distinctions in Cannabis adaptation strategies.

Response: This is a contested area, what is truly wild, what is feral, and what is wild crafted. This is why we reduced the number of occurrence points to choose those that showed no indication of human intervention.

Comment: The authors used 137 occurrence points to model habitat suitability globally, which may be a limited representation, especially given the scale of the analysis. A discussion addressing whether this sample size sufficiently captures global suitability would be valuable, perhaps mentioning potential limitations and the implications of using Eurasian data exclusively for a global SDM.

Response: We have added the statement

“Further, the limited sample size of the wild occurrences may lead to increased variability.”

As for the limitation of using Eurasian samples, this was done as this is the native range for Cannabis. There are strong reasons for only using these points to explore future niches. By capturing the climate variability that the species experiences in its native range here, we can then predict suitability in other non-observed areas (such as the American continent) where climatic properties match. This therefore does not limit the potential or translation to a global SDM model when only using Eurasian data points.

Comment: Although the paper explores past and present distribution shifts, it lacks a focused discussion on historical changes in the distribution of Cannabis. Integrating an analysis or discussion of historical fluctuations in its range, especially in response to past climatic shifts, could provide a more comprehensive context.

Response: This would be quite interesting but is not the major goal of this paper. The major goal here is to explore present day potential species ranges and how potential future changes may impact cultivation areas. It is difficult to translate the range changes in the past to the effects these may have had on the evolutionary trajectory of the plant without matched DNA or pollen samples for those specific timepoints. Work by Prof John McPartland has covered an exploration of past niches with matching pollen samples and was referenced here in this publication and in discussion.

Comment: While the authors effectively discuss how climate variables have influenced Cannabis distribution, soil properties such as organic carbon content and pH also vary over millennia. A brief discussion on how soil characteristics have changed historically could help contextualize the model’s results, as climate and soil variability are interrelated in shaping Cannabis habitats.

Response: We feel that this is outside the scope of the current paper to discuss historical biogeochemical cycling in soils, as the main goal was to explore the present day and potential future ranges for the species with respect to climate. We only included the soil data for the present day data analysis as we agree, soil properties are subject to change and much more difficult to rely upon through large timeframes at such broad resolutions.

Reviewer 4

Comment:The manuscript, titled "Species Distribution of Cannabis sativa: Past, Present and Future," offers valuable insights into how this species’ distribution responds to climate change. By modeling the historical, current, and projected future ranges of Cannabis sativa, the study highlights key environmental factors that shape its distribution under different climate scenarios. This research adds meaningfully to our understanding of the expansion-contraction model, especially for species affected by environmental shifts.

Response: thank you for the summary.

Comment:In terms of modeling, the use of AUC as a performance measure is helpful; however, including AICc and ROC values would make the model evaluation even more robust. AICc would allow for a clearer comparison of model fit across alternatives, while the ROC curve would provide a more complete view of predictive performance. These additions could be especially useful for readers with an interest in model selection and accuracy.

Response: This is a good point. We apologize but we have not included the AIC as we no longer have access to the model output and because we did not use them to make our model decisions. The ROCs for each analysis are available in the supplementary files.

Comment: Additionally, the discussion on expansion and contraction might benefit from referencing the study by Ülker, Tavşanoğlu, and Perktaş (2018) on Quercus robur ("Ecological niche modeling of Pedunculate Oak supports the ‘Expansion-Contraction’ model of Pleistocene biogeography"). Including this work may offer a useful comparison and add depth to the discussion of species distribution changes over time.

Response: we have added the sentence:

“Similar to previous work we see an expansion and contraction at known times of historical climate fluctuations (Ulker et al., 2018).”

Ülker, E. D., Tavşanoğlu, Ç., & Perktaş, U. (2018). Ecological niche modelling of pedunculate oak (Quercus robur) supports the ‘expansion–contraction’model of Pleistocene biogeography. Biological Journal of the Linnean Society, 123(2), 338-347.

Comment: Finally, it’s clear that the authors have made considerable improvements based on previous feedback, especially in addressing the limitations of soil data, explaining data selection, and refining methodological details. These thoughtful updates have successfully resolved earlier concerns and have strengthened the manuscript overall.

Response: Thank you

Reviewer 5

Comment: All comments have been addressed

Response: Thank you for looking at the specific comments we were asked to address instead of bringing up different new tangential concerns

Comment: In this study, authors have used various multi-source spatial and non-spatial data to investigate species Distribution of Cannabis sativa for Past, Present and future. It is an impressive compilation of information related with the documentation of Cannabis sativa.

Response: Thanks you for the summary

Comment: Authors have used CMIP5 GCM dataset for future period. Cannabis are sensitive to temperature and humidity. Previous CMIP5 models have higher uncertainty in temperature and rainfall than CMIP6 GCMs. I suggest applying a dataset of CMIP6 GCMs, otherwise provide uncertainty in estimated suitability areas due to the use of CMIP5 GCMs.

Response: We apologize if this was unclear, but this is incorrect, we stated in the methods we used CMIP6

---

## [Decision Letter · Decision Letter 2]

28 Nov 2024

Species Distribution of  Cannabis sativa : Past, Present and future

PONE-D-24-23287R2

Dear Dr. Kantar,

We’re pleased to inform you that your manuscript has been judged scientifically suitable for publication and will be formally accepted for publication once it meets all outstanding technical requirements.

Kind regards,

Andrea Mastinu

Academic Editor

PLOS ONE

Additional Editor Comments (optional):

Reviewers' comments:

Reviewer's Responses to Questions

**Comments to the Author**

1. If the authors have adequately addressed your comments raised in a previous round of review and you feel that this manuscript is now acceptable for publication, you may indicate that here to bypass the “Comments to the Author” section, enter your conflict of interest statement in the “Confidential to Editor” section, and submit your "Accept" recommendation.

Reviewer #2: (No Response)

Reviewer #4: All comments have been addressed

2. Is the manuscript technically sound, and do the data support the conclusions?

Reviewer #2: Yes

Reviewer #4: Yes

3. Has the statistical analysis been performed appropriately and rigorously? 

Reviewer #2: Yes

Reviewer #4: Yes

4. Have the authors made all data underlying the findings in their manuscript fully available?

Reviewer #2: Yes

Reviewer #4: Yes

5. Is the manuscript presented in an intelligible fashion and written in standard English?

Reviewer #2: Yes

Reviewer #4: Yes

6. Review Comments to the Author

Reviewer #2: The authors have sufficiently addressed my concerns. The manuscript now has improved in comparison to the previous version

Best wishes,

Reviewer #4: Thank you for addressing my comments effectively. I extend my congratulations on the nice execution of this work.

7. PLOS authors have the option to publish the peer review history of their article (what does this mean? ). If published, this will include your full peer review and any attached files.

**Do you want your identity to be public for this peer review?** For information about this choice, including consent withdrawal, please see our Privacy Policy .

Reviewer #2: No

Reviewer #4: No

---

## [Editor Report · Acceptance letter]

PONE-D-24-23287R2

PLOS ONE

Dear Dr. Kantar,

I'm pleased to inform you that your manuscript has been deemed suitable for publication in PLOS ONE. Congratulations! Your manuscript is now being handed over to our production team.

Kind regards,

on behalf of

Dr. Andrea Mastinu

Academic Editor

PLOS ONE